# Utility of Artificial Intelligence for Decision Making in Thoracic Multidisciplinary Tumor Boards

**DOI:** 10.3390/jcm14020399

**Published:** 2025-01-10

**Authors:** Jon Zabaleta, Borja Aguinagalde, Iker Lopez, Arantza Fernandez-Monge, Jon A. Lizarbe, Maria Mainer, Juan A. Ferrer-Bonsoms, Mateo de Assas

**Affiliations:** 1Department of Thoracic Surgery, Basque Health Service, Donostialdea Integrated Health Organisation, 20014 San Sebastian, Spain; borja.aguinagaldevaliente@osakidetza.eus (B.A.);; 2Tecnun, School of Engineering, University of Navarra, 20018 San Sebastian, Spain

**Keywords:** artificial intelligence (AI), lung cancer, multidisciplinary tumor board, decision making

## Abstract

**Background/Objectives**: The aim of this study was to analyze whether the implementation of artificial intelligence (AI), specifically the Natural Language Processing (NLP) branch developed by OpenAI, could help a thoracic multidisciplinary tumor board (MTB) make decisions if provided with all of the patient data presented to the committee and supported by accepted clinical practice guidelines. **Methods**: This is a retrospective comparative study. The inclusion criteria were defined as all patients who presented at the thoracic MTB with a suspicious or first diagnosis of non-small-cell lung cancer between January 2023 and June 2023. Intervention: GPT 3.5 turbo chat was used, providing the clinical case summary presented in committee proceedings and the latest SEPAR lung cancer treatment guidelines. The application was asked to issue one of the following recommendations: follow-up, surgery, chemotherapy, radiotherapy, or chemoradiotherapy. Statistical analysis: A concordance analysis was performed by measuring the Kappa coefficient to evaluate the consistency between the results of the AI and the committee’s decision. **Results**: Fifty-two patients were included in the study. The AI had an overall concordance of 76%, with a Kappa index of 0.59 and a consistency and replicability of 92.3% for the patients in whom it recommended surgery (after repeating the cases four times). **Conclusions**: AI is an interesting tool which could help in decision making in MTBs.

## 1. Introduction

Lung cancer is the leading cause of cancer-related mortality in the Western world [1]. Although the creation and presentation of cases in multidisciplinary tumor boards (MTBs) to decide on the most appropriate intervention in patients with lung cancer have shown improved survival outcomes [2], not all hospitals have access to such boards.

Meanwhile, the integration of artificial intelligence (AI) into healthcare decision making has emerged as a promising field for enhancing the accuracy and efficiency in disease management [3]. This represents a significant advancement in personalized medicine, as these technologies, powered by large volumes of data, enable medical decisions to be tailored with a much greater precision to each individual patient. In previous studies, the application of deep learning algorithms has demonstrated utility in identifying complex patterns in medical images, such as chest CT scans, thereby facilitating the early detection and accurate classification of lung lesions [3,4]. The capability of these tools to analyze large radiological datasets has led to significant improvements in identifying malignant lung nodules, allowing for quicker and more accurate diagnoses. Beyond early detection, AI has also proven valuable in prognostic prediction [5] and in personalizing oncology treatments [6]. Recent research has explored AI models that integrate clinical, genetic, and histopathological data to predict treatment responses and optimize therapeutic strategies [7,8]. These approaches enable more informed and customized decision making, tailored to the specific characteristics of each patient, thereby improving the overall effectiveness of treatment protocols [9].

The primary tool utilized in this project was ChatGPT. This tool is an artificial intelligence system capable of understanding and generating meaningful text, a field known as Natural Language Processing (NLP). NLP is a branch of artificial intelligence focused on the interaction between computers and human language. Its goal is to enable computers to understand and generate human language in a useful and meaningful way [10]. NLP encompasses tasks ranging from basic text analysis to complex language comprehension. It can be broadly categorized into the following two key areas [10]:

Natural Language Understanding (NLU) and Natural Language Generation (NLG). NLU allows machines to comprehend natural language by analyzing it and extracting key concepts, entities, emotions, and keywords. NLG, on the other hand, involves generating phrases, sentences, and paragraphs that convey meaningful and coherent information.

A subcategory of NLP is large language models (LLMs), which are responsible for advanced text comprehension and generation tasks. Compared to traditional NLP systems, LLMs are significantly more complex, requiring vast amounts of data and computational power to function effectively [10]. Their capacity for both understanding and generating human language at an advanced level has made them a cornerstone of modern NLP applications, including their use in medical decision support systems like the one explored in this study.

In 2024, Nardone et al. published an interesting review that explores the perspectives of clinicians engaged in the care of cancer patients, including surgeons, medical oncologists, and radiation oncologists, on the utilization of AI in MTBs [11]. Furthermore, the role of AI in various clinical specialties involved in the diagnosis and treatment of cancer was examined. By analyzing both the potential and the challenges, this study demonstrates how AI can facilitate enhanced multidisciplinary discourse and optimize treatment plans. The results demonstrate the capacity of AI to facilitate transformative improvements in cancer care and to maintain the efficacy of MTBs in the context of mounting clinical demands [11]. If we look more concretely at the possibility of using AI in decision making, similar to what we are considering in this paper, previous work has tested different AIs in the context of MTBs with breast cancer patients [12,13] with promising results. In our study, we evaluate whether ChatGPT could replicate the decision-making process of a multidisciplinary tumor board (MTB) for patients with confirmed or suspected lung cancer when provided with a clinical case summary and the SEPAR lung cancer treatment guidelines [1].

## 2. Materials and Methods

This is a retrospective study in which data were collected from patients presented at the tumor board between January 2023 and April 2023.

The summary presented at the tumor board was recorded as free text. It included the patient’s personal medical history, standard treatments, and reports of all complementary tests performed to date. The SEPAR lung cancer treatment guidelines were provided to the AI to enable decision making based on these guidelines. The AI application was asked to make a recommendation for each case from the following five possible options: follow-up, surgery, chemotherapy, radiotherapy, or chemoradiotherapy.

### 2.1. Study Subjects

The study population consisted of all patients who presented at the thoracic tumor board during the study period and that were correctly scheduled on the agenda. In order to be included in the study, patients were required to meet the inclusion criteria and not to meet any of the exclusion criteria. The inclusion criteria were as follows: patients with an initial diagnosis or suspicion of lung cancer that presented at the tumor board with the objective of determining the subsequent steps to be taken for the patient in question. Patients were excluded from the study if they met any of the following criteria: patients who presented to the tumor board for radiological evaluation only, patients with tumors of other origins presenting with lung metastases, and patients previously treated for lung cancer and referred to assess recurrence.

### 2.2. Variables

The variables collected for subsequent statistical analysis were as follows: confirmation of the histological diagnosis at the time of presentation of the case to the board (collected as a dichotomous variable: Yes/No); in the summary presented to the board, the clinical TNM of the patient appears (collected as a dichotomous variable: Yes/No); all of the patient information presented at the committee was collected as free text (personal history, usual treatment, and complementary tests); how many times the case was presented at the MTB (collected as a continuous variable) and the decision taken by the MTB (follow-up, surgery, chemotherapy, radiotherapy, or chemoradiotherapy); the outcome variable analyzed was the recommendation offered by the AI (follow-up, surgery, chemotherapy, radiotherapy, or chemoradiotherapy).

### 2.3. Statistical Analysis

For the descriptive analysis, the most appropriate statistical measure was used in each case: the mean and median for continuous variables, and absolute values and percentages for categorical variables. The chi-square test was employed to compare categorical variables, while the Kappa index was applied to assess interobserver agreement (observer 1 being the AI and the other being the MTB), measuring the degree of variability between the observers. We also analyzed the accuracy and recall to evaluate the performance of a predictive model. Accuracy measures the proportion of true positives among all of the instances predicted as positive (i.e., when the AI predicts A, the probability that the committee’s result is also A). Recall, on the other hand, quantifies the ratio of correctly predicted positives to the total number of actual positives (i.e., how well the AI performed against the gold standard). To evaluate the study’s replicability, precision was measured in addition to accuracy. Precision indicates the number of times the most frequently predicted option was chosen, as follows:Precision (%) = Number of times the most frequent option is predicted/Number of repetitions × 100.

The cases in which the AI achieved 100% precision but an accuracy of zero were analyzed in detail to identify the factors that influenced the AI’s incorrect decision.

### 2.4. Tools

The study was conducted using the ChatGPT-3.5 turbo version of OpenAI (version 1.14.0). In order to elicit a response from the AI, the following prompt was generated: “You will make decisions typically made in a lung tumor board (and we add the SEPAR guideline in a pdf file). Patient information will now be provided (and we add all the information of the patient). Using the SEPAR guideline and the patient information, please recommend only one of the following options. There are five options: Surgery, Radiotherapy, Chemotherapy, Chemotherapy and Radiotherapy, Follow-up”.

### 2.5. Patient Management

At the tumor board, the case was presented with all of the following patient data included in the summary: personal medical history, standard treatment, and reports of all complementary tests performed to date. Additionally, CT scans and PET/CT images were reviewed with radiologists and nuclear medicine specialists. The tumor board involved pulmonologists, thoracic surgeons, medical oncologists, and radiation oncologists, and, for each patient, one of the following options was decided: surgery, radiotherapy, chemotherapy, chemotherapy + radiotherapy, or follow-up. Once the approach was determined, the patient was referred to the appropriate specialist’s consultation.

## 3. Results

The study included a total of 52 patients. In 19 instances (36.5%), the TNM classification was not documented, and, in 28 cases (53.8%), no histological diagnosis was available at the time of presentation to the tumor board. In 27 cases (51.9%), the patient was being presented to the MTB for the first time, while, in 25 cases (48.1%), it was the second or third presentation. When raw data were provided to the AI, the overall concordance was 63%. Upon incorporating the TNM information, concordance increased to 75%. Therefore, all of the subsequent analyses were conducted using data that included the TNM information (Table 1).

An analysis was conducted to determine whether the presence of histological data at the time of presentation affected the concordance. The concordance for patients without histology was 75% (21 correct recommendations out of 28 cases) versus 75% for those with histology (18 out of 24 cases) (*p* = 1).

When considering whether the MTB’s decision was made during the patient’s first presentation, the following results were observed: the concordance for patients presented to the committee for the first time was 70.37% (19 correct recommendations out of 27 cases), while, for cases decided upon after multiple presentations, the concordance reached 80% (20 out of 25 cases) (*p* = 0.423).

Table 2 presents the confusion matrix for the 52 cases, showing the AI’s recommendations based on the MTB’s decision. The Kappa index for the AI-based decisions was 0.59.

To assess the replicability of the AI, accuracy and precision metrics were calculated by repeating the same cases in a different order across four iterations. Table 3 presents the AI’s accuracy and precision, broken down by the decision type, over these four repetitions.

### Special Cases

In the analysis of the replicability test results, several cases showed a high precision (100%) but with an accuracy of zero—meaning that the AI consistently made the same recommendation, but it differed from the MTB’s decision.

Special Case 1: The AI recommended surgery, but radiotherapy was ultimately chosen. The case notes indicated that “the patient was tired of undergoing tests and did not wish for much more”. Hence, radiotherapy was selected as the therapeutic option. A more appropriate approach might have been to state that surgery or radiotherapy could both be suitable, contingent on patient preference.Special Case 2: Similar to Case 1, the AI recommended surgery, while radiotherapy was chosen by the MTB. Although surgery was ideal based on the tumor characteristics, the patient’s age (82), poor respiratory function test results, and diagnosis amid acute respiratory failure led the committee to opt for SBRT. In this case, the patient’s overall assessment weighed more heavily than the tumor characteristics in the decision-making process.Special Case 3: The patient had two lung lesions, but the MTB considered one benign and based its decision on this assessment. The AI, however, interpreted the lesion as malignant due to incomplete information in the summary prior to the MTB meeting. The AI recommended surgery, while the committee opted for follow-up.

## 4. Discussion

The primary tool used in this project was ChatGPT [14], an artificial intelligence system capable of understanding and generating coherent text, a process known as Natural Language Processing (NLP). ChatGPT was developed by OpenAI, a research organization dedicated to advancing AI in a way that benefits humanity as a whole [14].

AI has been applied across various scenarios in respiratory health [15,16,17,18] with different objectives. Izquierdo et al. published the results of using AI software (Savana Manager, version 3.0) that enabled the extraction of data from electronic health records (EHRs), addressing a long-standing question posed by clinicians: how to efficiently obtain and analyze data from unstructured free-text entries in EHRs. Savana^®^ (Madrid, Spain) developed EHRead technology, which allows for the reading, processing, and organizing of non-structured free text from EHRs. Once this process is completed, the information from the EHRs is converted into structured data, which can then be easily and rapidly stored, consulted, and analyzed for research purposes. Using this software, they analyzed data related to asthma management and the use of systemic corticosteroids [15], as well as data from over 50,000 COPD patients over a 40-year period [15,16]. AI has also been used as a tool for providing rapid data analysis in COVID-19 cases, generating treatment, diagnostic, and prognostic recommendations [17]. Thanks to AI, they analyzed the data from 31,633 COPD patients, 793 of whom had a diagnosis of COVID-19. After analyzing admissions, complications, and mortality, they concluded that COVID-19 had a clinical profile different from exacerbations caused by other respiratory viruses during the winter season [17]. Exploring various uses of AI in respiratory health, a particularly interesting 2023 study investigated whether an AI model (specifically, ChatGPT) could pass a thoracic surgery board examination [18]. In this case, ChatGPT was given the 150 questions from the competitive exam for the post of Specialist in thoracic surgery announced by the Andalusian Health Service in 2022. The AI correctly answered 58.9% of the questions, with an accuracy rate of 86% for the questions answered correctly [18]. The threshold for passing, as specified in the official exam guidelines, was 60% of the average of the best 10 scores. A passing mark of 40 points was set, and the AI model would have passed this portion of the examination for a thoracic surgery physician/specialist with a score of 45.79 points [18].

In Spain, as in Europe and the United States, lung cancer remains one of the leading causes of cancer mortality, with over 30,000 new lung cancer cases (LC) diagnosed in 2022 [19]. Managing these patients is complex for numerous reasons, and a broad team of specialists is involved throughout diagnosis, treatment, and follow-up, including pulmonologists, thoracic surgeons, medical oncologists, radiation oncologists, radiologists, pathologists, and nurse case managers, respectively; multidisciplinary tumor boards are essential, as making patient-related decisions in a committee enhances the adherence to clinical practice guidelines and improves survival rates [2,20,21]. Recently, in Spain, standards of excellence and quality indicators for tumor boards have been proposed, along with recommendations for accreditation and auditing [22,23]. In line with these projects, this retrospective study can be used to analyze some of the quality data considered for accreditation that stand out, such as the absence of histology prior to intervention or the absence of TNM staging in the summary. Recently, our team published a study [24] analyzing the results of different strategies (more or less aggressive approaches to obtaining histological diagnosis before surgery) across four hospitals. The conclusion of this study was that the rate of histological sampling before lung cancer surgery still varies between hospitals. Despite very diverse multidisciplinary management, the rate of futile lobectomy is not significantly higher in hospitals with lower rates of preoperative histological analysis [24]. The absence of TNM staging in 36% of the cases was surprising when reviewing the data. Though clinical TNM staging was discussed for each case in the tumor board to make the decision, it was not reflected in the patient’s medical history. This retrospective work has, among other things, helped to improve this aspect of the tumor board, and now the TNM staging is documented for 100% of the cases.

In this challenging environment of striving for excellence, we sought to explore AI’s utility as a tool to support clinical decision making (CDS). Indeed, AI has emerged as a powerful CDS support tool. A systematic review published in 2024 concluded that AI is beneficial across the following six CDS domains: Data-Driven Insights and Analytics, Diagnostic and Predictive Modeling, Treatment Optimization and Personalized Medicine, Patient Monitoring and Telehealth Integration, Workflow and Administrative Efficiency, and Knowledge Management and Decision Support [25]. The review also concluded that “AI is revolutionizing healthcare by enhancing CDS in several domains, contributing to more efficient, effective, and patient-centric care. However, it should complement, not replace, human expertise” [25].

Our experience using ChatGPT as a decision-making aid in the MTB aligns with this perspective; we believe AI could summarize substantial patient data, analyze current guidelines, and provide recommendations for clinicians, with the final decision still being clinician-led. It could also be useful for smaller or remote centers that lack an MTB, helping the responsible physician refer the patient to the appropriate specialist and potentially reducing unnecessary travel or avoidable delays.

In 2023, Allowais et al. published a comprehensive review on the use of artificial intelligence (AI) in medicine titled “Revolutionizing healthcare: the role of artificial intelligence in clinical practice” [26]. The authors stated that integrating AI into healthcare holds significant potential for enhancing disease diagnosis, treatment selection, and clinical laboratory testing. AI tools can leverage large datasets and identify patterns that surpass human performance in various healthcare domains. AI offers increased accuracy, cost reduction, and time efficiency while minimizing human errors. It has the potential to revolutionize personalized medicine, optimize medication dosages, enhance population health management, establish clinical guidelines, provide virtual health assistants, support mental health care, improve patient education, and influence patient–physician trust. Despite its promise, AI is still in the early stages of being fully utilized for medical diagnosis. However, growing evidence supports the application of AI in diagnosing various diseases, including cancer [26]. Focusing on the use of AI in lung cancer in recent years, several studies have targeted different stages of the diagnostic and treatment process. For early diagnosis, Wang et al. utilized deep learning to improve low-dose CT (LDCT) screening [27], where a total of 3326 patients who underwent LDCT for lung cancer screening with subsequent pathological confirmation of lung cancer were included. Their deep learning approach (integrating radiomics and patient demographics) reduced the false-positive rate from 0.41 ± 0.02 to 0.30 ± 0.12, maintained the same false-negative rate, and improved the overall diagnostic performance [27]. Similarly, Heidari et al. [28] worked with deep learning (DL) and blockchain technology to address one of the main challenges—preserving data privacy during data sharing. In this study, the authors proposed an approach that utilized a modest amount of data from multiple hospitals and employed blockchain-based Federated Learning (FL) to train a global DL model. Blockchain technology authenticated the data, and FL trained the model on an international scale while maintaining organizational anonymity. The proposed method was trained and tested on datasets including the Cancer Imaging Archive, Kaggle Data Science Bowl, LUNA 16, and a local dataset. The results demonstrated an effective detection of lung cancer patients, achieving an accuracy of 99.69% [28]. To assess the treatment response, a systematic review and meta-analysis evaluated the utility of AI in predicting radiotherapy outcomes [29], and, in 2024, another study aimed to predict the impact of chemotherapy on cancer cells [30]. The aforementioned meta-analysis [29] included 18 studies with a total of 4719 patients, in which AI models were used to predict the survival outcomes in lung cancer patients following radiotherapy. The study demonstrated the clinical feasibility of using AI models to predict post-radiotherapy outcomes in lung cancer patients [29]. Regarding chemotherapy, deep neural network models have been utilized to predict the impact of anticancer drugs on tumors through the half-maximal inhibitory concentration (IC50). These models integrate biological and chemical data to extract relevant features from both genetic profiles and drug compounds. To predict the drug response in cancer cell lines, the study employed various deep learning approaches, including Recurrent Neural Networks (RNNs) and Convolutional Neural Networks (CNNs) [30].

AI has also been used as a decision support tool for cancer patients. Oncology currently has the broadest application of AI in medical research, with 50% of publications from 2017 to 2021 related to cancer fields. As of May 2021, 71 AI-based devices have received FDA clearance for clinical use, mostly in tumor radiology (55%) and pathology (20%) [11]. Similar studies to ours have been conducted with breast cancer patients: one study used ChatGPT to support individualized and personalized therapy, but found that “the current version is not able to provide specific recommendations for the therapy of patients with primary breast cancer” [31]. Another study using ChatGPT 3.5 found “overall concordance between the LLM and MTB is reached for half of the patient profiles, including precancerous lesions, and for invasive breast cancer profiles, concordance amounts to 58.8%” [12]. In a recent 2024 study, five LLMs—including three versions of ChatGPT (versions 4 and 3.5, with data up to September 2021 and January 2022), Llama2, and Bard—were tasked with generating treatment recommendations for 20 complex breast cancer cases. GPT-4 demonstrated the highest concordance (70.6%) for invasive breast cancer cases, followed by GPT-3.5 September 2021 (58.8%), GPT-3.5 January 2022 (41.2%), Llama2 (35.3%), and Bard (23.5%) [13]. In our study, we achieved better concordance, likely due to incorporating a clinical practice guideline [1] in the prompt. We believe future improvements in concordance could build on previous findings, as the scientific community plays an active role in ensuring the accuracy and validity of the information generated by AI models, the data used in training, and the evaluation of these tools for medical applications [32].

### 4.1. Limitations

This was an exploratory study with a small patient sample, under specific presentation conditions in the tumor board. To generalize these results, we consider it necessary to conduct a multicenter, prospective study, where each presented case includes a minimum dataset. Additionally, the prompt should include a current, widely accepted clinical practice guideline across all participating hospitals. Prompt design is a critical factor in optimizing the performance of AI systems like ChatGPT, especially in the context of tumor board decision making. By carefully structuring the prompts, clinicians can ensure that AI outputs are highly relevant, context-specific, and aligned with evidence-based guidelines. For instance, detailed prompts that include patient-specific variables—such as the tumor type, stage, genetic mutations, and previous treatment history—can guide the AI to produce recommendations tailored to the clinical scenario. Furthermore, standardizing the prompts across cases can enhance the consistency, enabling tumor boards to compare AI-generated suggestions more reliably. Effective prompt design also reduces ambiguity, minimizing the risk of irrelevant or incomplete outputs, and fosters the incorporation of up-to-date clinical guidelines by explicitly directing the AI to reference these sources. Overall, improved prompt design has the potential to significantly augment the utility of AI in supporting precise, informed, and reproducible responses. Prompt design is not the only critical element for enhancing the performance of AI systems like ChatGPT in clinical decision making; fine-tuning also plays a pivotal role. Fine-tuning involves adjusting a pre-trained model by incorporating domain-specific data, in this case, clinical data related to oncology. This process enables the AI to better understand the nuances of medical cases and generate more precise responses tailored to the clinical context and specific treatment guidelines. For instance, a model fine-tuned with data on different tumor types, genetic variants, and treatment responses could provide more personalized and evidence-based recommendations for real-world patients. Additionally, fine-tuning can integrate updated knowledge from biomedical databases, scientific literature, and authoritative clinical guidelines, such as those from SEPAR, NCCN, or ESMO. In the long term, fine-tuning also offers the potential to incorporate local and multicenter datasets to better reflect the characteristics of specific populations, which is particularly valuable in regions where cancer epidemiology or therapeutic resources differ from global standards. This ensures that AI-generated recommendations are both relevant and applicable to diverse clinical settings. When combined with advances in prompt design and technologies such as Federated Learning to maintain data privacy, fine-tuning can transform AI into an even more powerful tool, capable of complementing and optimizing tumor board decision making while improving the quality of care and patient outcomes.

### 4.2. Ethical Considerations and Implementation Challenges

The integration of AI systems like ChatGPT into tumor board decision making raises several ethical and practical considerations that warrant careful examination. One of the primary ethical concerns is the potential for AI to reinforce biases present in training datasets. If not appropriately managed, these biases could lead to inequities in treatment recommendations, particularly for underrepresented populations. Moreover, the lack of transparency in how AI models generate their outputs—the so-called “black box” problem—may undermine clinicians’ trust and pose challenges in ensuring accountability for decisions influenced by AI.

Another critical consideration is data privacy and security. AI systems require access to sensitive patient data, which must be managed in compliance with stringent regulations. While technologies like Federated Learning and blockchain offer promising solutions (as described by Heidari et al. [28]), their implementation in real-world clinical settings remains complex and resource-intensive.

From an implementation perspective, integrating AI tools into clinical workflows requires significant adjustments, including clinician training and updates to existing health information systems. Ensuring that clinicians understand the limitations of AI recommendations and use them as complementary to, rather than a replacement for, expert judgment is paramount. Additionally, regulatory frameworks for the validation and approval of AI tools in healthcare must be addressed to ensure safety and reliability before widespread deployment.

These limitations highlight the need for a cautious, multidisciplinary approach to adopting AI in clinical practice, involving ethicists, data scientists, and healthcare professionals to maximize the benefits while mitigating the risks.

## 5. Conclusions

AI could serve as a valuable tool for clinical decision making in a multidisciplinary lung tumor board, though further studies and process improvements are needed before it can be considered as a reliable tool.

## Figures and Tables

**Table 1 jcm-14-00399-t001:** Comparison of cases with and without the TNM.

AI Recommendation	TNM Not Described	TNM Described	Total
Incorrect	10	3	13
Correct	9	30	39

Pearson’s chi-square test; *p* < 0.001.

**Table 2 jcm-14-00399-t002:** Confusion matrix.

	Artificial Intelligence Decision					Recall
MTB Decision	Surgery	QT+RT	QT	RT	Follow-Up	
Surgery	25	1	0	0	0	96.15%
QT+RT	4	8	0	0	0	66.7%
QT	1	1	2	0	0	50.0%
RT	2	0	1	1	0	25.0%
Follow-up	2	0	0	0	2	50.0%
Accuracy	73.5%	80.0%	66.7%	100%	100%	

QT, chemotherapy; RT, radiotherapy.

**Table 3 jcm-14-00399-t003:** AI accuracy and precision by decision after 4 repetitions.

AI Decision	Accuracy	Precision
Surgery	92.3%	92.3%
Chemo+RT	71.6%	60%
Chemotherapy	95%	45%
Radiotherapy	80%	30%
Follow-up	80%	25%

## Data Availability

The data supporting the findings of this study are available upon request from the corresponding author.

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
