# Peer review of "Utility of Artificial Intelligence for Decision Making in Thoracic Multidisciplinary Tumor Boards"

_jcm, 2025, doi:10.3390/jcm14020399_

Round 1
Reviewer 1 Report
Comments and Suggestions for Authors
Dear authors, the manuscript that you submit to Journal of Clinical Medicine deals with an interesting and actual topic.
The study design is retrospective and based on a comparison between a multidisciplinary decision and a AI (chat GPT) decision concerning treatment choice for patients affected by lung cancer.
The findings are suitable to be shared and debated , however many issues reamins open:
1) the population is limited
2) an external validation (with another multidisciplinary board) is missing
3) the information provided for each case to Chat GPT is limited and lower than commonly available in the clinical practice during the MDT discussion.
4) the authors conclude that the corrispondance between MDT eand AI is good, but tthe level of accordance/agreement expected is not declared . It could be important in order to confirm these findings in a prospective and larger cohort.
These points should be discussed and improved in the text.
Moreover, an ethical consideration of the approach with AI in the decision making and a delineation of the possible implementation inthe clinical practice would be appreciated.
Author Response
Thank you very much to the editor and reviewers for taking the time to read and analyze the article we have prepared. We believe that the responses to the comments and questions raised by the two reviewers have greatly contributed, and thanks to this, the reader's understanding will be enhanced, and we have achieved a higher-quality manuscript. Below, we detail the responses to each comment.
Reviewer 1
Dear authors, the manuscript that you submit to Journal of Clinical Medicine deals with an interesting and actual topic.
The study design is retrospective and based on a comparison between a multidisciplinary decision and a AI (chat GPT) decision concerning treatment choice for patients affected by lung cancer.
The findings are suitable to be shared and debated , however many issues reamins open:
1) the population is limited
2) an external validation (with another multidisciplinary board) is missing
Response to comments 1 and 2: It is true that the study population is limited. After retrospectively verifying that this tool can be useful, the next step will be to conduct a prospective and multicenter study with a larger patient cohort. We have explicitly mentioned this point in the study limitations.
3) the information provided for each case to Chat GPT is limited and lower than commonly available in the clinical practice during the MDT discussion.
The information provided to ChatGPT is the same as that presented to the tumor board for decision-making. To ensure clarity, we have modified the Materials and Methods section to make this point more explicit adding: “The summary presented at the tumor board was recorded as free text. It included the patient's personal medical history, standard treatments, and reports of all complementary tests performed to date.”
4) the authors conclude that the corrispondance between MDT eand AI is good, but tthe level of accordance/agreement expected is not declared . It could be important in order to confirm these findings in a prospective and larger cohort.
We state that the accordance is good because we have improved the results of previous studies, where concordance rates reached up to 70%, and because the kappa index achieved was 0.59. We consider this work to be a solid starting point for improving concordance in future studies, as mentioned in the discussion. “We believe future improvements in concordance could build on previous findings, as the scientific community plays an active role in ensuring the accuracy and validity of the information generated by AI models, the data used in training, and the evaluation of these tools for medical applications”
These points should be discussed and improved in the text.
Moreover, an ethical consideration of the approach with AI in the decision making and a delineation of the possible implementation in the clinical practice would be appreciated.
We have added the following paragraphs after limitations section.
Ethical Considerations and Implementation Challenges
The integration of AI systems like ChatGPT into tumor board decision-making raises several ethical and practical considerations that warrant careful examination. One of the primary ethical concerns is the potential for AI to reinforce biases present in training datasets. If not appropriately managed, these biases could lead to inequities in treatment recommendations, particularly for underrepresented populations. Moreover, the lack of transparency in how AI models generate their outputs—the so-called "black box" problem—may undermine clinicians' trust and pose challenges in ensuring accountability for decisions influenced by AI.
Another critical consideration is data privacy and security. AI systems require access to sensitive patient data, which must be managed in compliance with stringent regulations. While technologies like federated learning and blockchain offer promising solutions (as described by Heidari et al [5]), their implementation in real-world clinical settings remains complex and resource-intensive.
From an implementation perspective, integrating AI tools into clinical workflows requires significant adjustments, including clinician training and updates to existing health information systems. Ensuring that clinicians understand the limitations of AI recommendations and use them as complementary to, rather than a replacement for, expert judgment is paramount. Additionally, regulatory frameworks for the validation and approval of AI tools in healthcare must be addressed to ensure safety and reliability before widespread deployment.
These limitations highlight the need for a cautious, multidisciplinary approach to adopting AI in clinical practice, involving ethicists, data scientists, and healthcare professionals to maximize benefits while mitigating risks

Reviewer 2 Report
Comments and Suggestions for Authors
This paper evaluates the use of AI (GPT-3.5 Turbo) to support decision-making in thoracic tumor boards for non-small cell lung cancer, achieving 76% concordance with human decisions. Its strength lies in integrating AI with clinical guidelines, demonstrating AI’s potential to assist in clinical settings with limited MTB access.
Abstract:
1. The sentence: "Inclusion criteria were all patients presented at thoracic MTB with first diagnosis of non-small cell lung cancer between January 2023 and June 2023 have been collected" requires grammatical correction.
Intro:
- Clarify the inconsistent use of black, red, and blue text formatting. Ensure uniformity.
- Specify the decision-making process in the tumor board:
- Are multiple treatment options discussed, or is only one chosen?
- If multiple options are discussed, is there a ranking system or prioritization of choices?
- Provide a broader overview of existing literature on AI in tumor boards for other cancer types. Several recent publications are available that could strengthen this context.
Materials and Methods:
- Lines 81–99: It is unclear if the content is part of a table or free text. Please clarify.
- Define the term "variables" explicitly
- Ensure consistent formatting for subheadings (e.g., italicization). Some are italicized, while others are not.
- The phrase: "IA and tumor board, being the MTB decision the gold standard" is unclear.
- Tools Section: Explain for clarity.
- For better clarity, rewrite sections describing "Prompt synthesis" in full sentences. Include coding details in supplementary material.
- Clarify again: Are multiple treatment options presented, or is only one recommendation given? Is there a ranking system?
Results:
- Address the inconsistent use of red text formatting.
- Discuss the significant amount of missing data:
- Histology missing (53.8%) and TNM staging missing (36.5%): How can tumor board recommendations be made without such critical data?
- Clarify how decisions were reached in these cases and discuss the limitations this poses.
- Special cases are presented in the Results but are not mentioned in Methods. These cases should be introduced as part of the methodology to maintain structure.
Discussion:
- In lines 255–230, potential AI applications are mentioned without corresponding results. Specific findings, especially from Reference 15, would strengthen the discussion.
- Incorporate more recent literature (post-2021) to reflect the current state of AI in oncology and tumor board applications.
- Elaborate on how prompt design could improve AI outcomes. What specific improvements are anticipated?
- Clarify this statement: "Additionally, the prompt should include a current, widely accepted clinical practice guideline across all participating hospitals, and a fine-tuning process should be applied."
- What does "fine-tuning" entail in this context? Is this a recommendation or part of future work?
General Comments:
- The manuscript is challenging to read due to frequent disruptions in text flow. Reorganize for a clearer, more continuous narrative.
- Improve grammar and sentence structure throughout the manuscript for better readability.
- Ensure consistency in formatting, including headings, text color, and subheadings.
Author Response
Thank you very much to the editor and reviewers for taking the time to read and analyze the article we have prepared. We believe that the responses to the comments and questions raised by the two reviewers have greatly contributed, and thanks to this, the reader's understanding will be enhanced, and we have achieved a higher-quality manuscript. Below, we detail the responses to each comment.
Reviewer 2
This paper evaluates the use of AI (GPT-3.5 Turbo) to support decision-making in thoracic tumor boards for non-small cell lung cancer, achieving 76% concordance with human decisions. Its strength lies in integrating AI with clinical guidelines, demonstrating AI’s potential to assist in clinical settings with limited MTB access.
Abstract:
- The sentence: "Inclusion criteria were all patients presented at thoracic MTB with first diagnosis of non-small cell lung cancer between January 2023 and June 2023 have been collected" requires grammatical correction.
Corrected, in the new version: “The inclusion criteria were defined as all patients who presented at the thoracic MTB with a suspicious or first diagnosis of non-small cell lung cancer between January 2023 and June 2023.”
Intro:
Clarify the inconsistent use of black, red, and blue text formatting. Ensure uniformity.
It was in different colors because the editor requested some changes before submitting the manuscript to the reviewers. Different colors were used to make the changes easy to identify. Everything is now in black.
Specify the decision-making process in the tumor board: Are multiple treatment options discussed, or is only one chosen? If multiple options are discussed, is there a ranking system or prioritization of choices?
As this is a retrospective study, the functioning of the MTB was not modified. In our center, the MTB makes a decision, and the patient is referred to the corresponding specialist with the committee's decision. To clarify this, we have added the following paragraph at the end of the Materials and Methods section.
Patient Management
At the tumor board, the case was presented with all the patient data included in the summary: personal medical history, standard treatment, and reports of all complementary tests performed to date. Additionally, CT scans and PET/CT images were reviewed with radiologists and nuclear medicine specialists. The tumor board involved pulmonologists, thoracic surgeons, medical oncologists, and radiation oncologists, and for each patient, one of the following options was decided: surgery, radiotherapy, chemotherapy, chemotherapy + radiotherapy, or follow-up. Once the approach was determined, the patient was referred to the appropriate specialist's consultation
Provide a broader overview of existing literature on AI in tumor boards for other cancer types. Several recent publications are available that could strengthen this context.
Following the suggestion, we added to the text:
“In 2024, Nardone et al. published an interesting review that explores the perspectives of clinicians engaged in the care of cancer patients, including surgeons, medical oncologists, and radiation oncologists, on the utilisation of AI in MTBs. Furthermore, the role of AI in various clinical specialities involved in the diagnosis and treatment of cancer is examined. By analysing both the potential and the challenges, this study demonstrates how AI can facilitate enhanced multidisciplinary discourse and optimise treatment plans. The results demonstrate the capacity of AI to facilitate transformative improvements in cancer care and maintain the efficacy of MTBs in the context of mounting clinical demands [11]. If we look more concretely at the possibility of using AI in decision making similar to what we are considering in this paper, previous work has tested different AIs in the context of MTB with breast cancer patients [12,13] with promising results. In our study, we evaluated whether ChatGPT could replicate the decision-making process of a multidisciplinary tumor board (MTB) for patients with confirmed or suspected lung cancer when provided with a clinical case summary and the SEPAR lung cancer treatment guidelines [1].”
Materials and Methods:
- Lines 81–99: It is unclear if the content is part of a table or free text. Please clarify.
Its a free text. We have revised it to improve the wording.
Study Subjects
The study population consisted of all patients who were presented at the thoracic tumour board during the study period and correctly scheduled on the agenda. In order to be included in the study, patients were required to meet the inclusion criteria and not to meet any of the exclusion criteria. Inclusion criteria: Patients with an initial diagnosis or suspicion of lung cancer were presented at the tumour board with the objective of determining the subsequent steps to be taken for the patient in question. Patients were excluded from the study if they met any of the following criteria: Patients who were presented for radiological evaluation only, patients with tumours of other origins presenting with lung metastases, and patients previously treated for lung cancer and referred to assess recurrence were excluded from the study.
- Define the term "variables" explicitly
We have expanded the paragraph, and the new version of the manuscript states the following: The variables collected for subsequent statistical analysis were as follows: confirmation of the histological diagnosis at the time of presentation of the case to the board (collected as a dichotomous variable: Yes/No); in the summary presented to the board, the clinical TNM of the patient appears (collected as a dichotomous variable: Yes/No); all patient information presented at the committee was collected as free text (personal history, usual treatment and complementary tests); how many times the case was presented at the MTB (collected as a continuous variable) and the decision taken by the MTB (follow-up, surgery, chemotherapy, radiotherapy, or chemoradiotherapy); the outcome variable analysed was the recommendation offered by the AI (follow-up, surgery, chemotherapy, radiotherapy, or chemoradiotherapy).
- Ensure consistent formatting for subheadings (e.g., italicization). Some are italicized, while others are not.
Corrected
- The phrase: "IA and tumor board, being the MTB decision the gold standard" is unclear.
We have removed that sentence and replaced it with this, which we believe is clearer: while the Kappa index was applied to assess interobserver agreement (observer 1 being the AI and the other the MTB),
- Tools Section: Explain for clarity.
- For better clarity, rewrite sections describing "Prompt synthesis" in full sentences. Include coding details in supplementary material.
In response to comments 5 and 6, we have revised this section and rewritten it as follows, which we believe is clearer:
Tools
The study was conducted using the ChatGPT-3.5 turbo version of OpenAI (version 1.14.0). In order to elicit a response from the AI, the following prompt was generated: “You will make decisions typically made in a lung tumor board (and we add the SEPAR guideline in a pdf file). Patient information will now be provided (and we add all the information of the patient). Using the SEPAR guideline and the patient information, please recommend only one of the following options. There are five options: Surgery, radiotherapy, Chemotherapy, Chemotherapy and Radiotherapy, Follow-up."
- Clarify again: Are multiple treatment options presented, or is only one recommendation given? Is there a ranking system?
Patient Management
At the tumor board, the case was presented with all the patient data included in the summary: personal medical history, standard treatment, and reports of all complementary tests performed to date. Additionally, CT scans and PET/CT images were reviewed with radiologists and nuclear medicine specialists. The tumor board involved pulmonologists, thoracic surgeons, medical oncologists, and radiation oncologists, and for each patient, one of the following options was decided: surgery, radiotherapy, chemotherapy, chemotherapy + radiotherapy, or follow-up. Once the approach was determined, the patient was referred to the appropriate specialist's consultation
Results:
- Address the inconsistent use of red text formatting.
As we mentioned previously, the use of two colors was to allow the editor to easily identify the changes from the first version. Everything is now in black.
- Discuss the significant amount of missing data: Histology missing (53.8%) and TNM staging missing (36.5%): How can tumor board recommendations be made without such critical data?
Clarify how decisions were reached in these cases and discuss the limitations this poses.
The absence of histology is because we often decide to proceed with surgery without a histological diagnosis when suspicion is high. Recently, our team published a study analyzing the results of different strategies (more or less aggressive approaches to obtaining a histological diagnosis before surgery) across four hospitals. The conclusion of that study was: The rate of histological sampling before lung cancer surgery still varies between hospitals. In spite of very diverse multidisciplinary management, the rate of futile lobectomy is not significantly higher in hospitals with lower rates of preoperative histological analysis. The absence of TNM staging in the records surprised us during the review: we understand that the clinical TNM staging was discussed for each case during the tumor board discussions but was not reflected in the patient's medical history. This retrospective study has helped, among other things, to improve this aspect of the tumor board process, and now the TNM staging is documented for 100% of cases.
In discusión section we added the next parragraph: In line with these projects, this retrospective study can be used to analyze some of the quality data considered for accreditation that stand out, such as the absence of histology prior to intervention or the absence of TNM staging in the summary. Recently, our team published a study [25] analyzing the results of different strategies (more or less aggressive approaches to obtaining histological diagnosis before surgery) across four hospitals. The conclusion of this study was that the rate of histological sampling before lung cancer surgery still varies between hospitals. Despite very diverse multidisciplinary management, the rate of futile lobectomy is not significantly higher in hospitals with lower rates of preoperative histological analysis [25]. The absence of TNM staging in 36% of the cases was surprising when reviewing the data: we understand that the clinical TNM staging was discussed for each case in the tumor board to make the decision, but it was not reflected in the patient's medical history. This retrospective work has, among other things, helped improve this aspect of the tumor board, and now the TNM staging is documented for 100% of the cases.
- Special cases are presented in the Results but are not mentioned in Methods. These cases should be introduced as part of the methodology to maintain structure.
The following has been added to the Methods section: Cases in which the AI achieved 100% precision but zero accuracy were analyzed in detail to identify factors influencing the AI's erroneous decision-making.
Discussion:
- In lines 255–230, potential AI applications are mentioned without corresponding results. Specific findings, especially from Reference 15, would strengthen the discussion.
We developed this first sentences and in the new versión we can read: AI has been applied across various scenarios in respiratory health [15-18] with different objectives. Izquierdo et al. published the results of using an AI software that enabled the extraction of data from electronic health records (EHRs), addressing a long-standing question posed by clinicians: how to efficiently obtain and analyze data from unstructured free-text entries in EHRs? Savana® (Madrid, Spain) developed EHRead technology, which allows for reading, processing, and organizing non-structured free text from EHRs. Once this process is completed, the information from the EHRs is converted into structured data, which can then be easily and rapidly stored, consulted, and analyzed for research purposes. Using this software, they analyzed data related to asthma management and the use of systemic corticosteroids [15], as well as data from over 50,000 COPD patients over a 40-year period [15,16]. AI has also been used as a tool for providing rapid data analysis in COVID-19 cases, generating treatment, diagnostic, and prognostic recommendations [17]. Thanks to AI, they analyzed data from 31,633 COPD patients, 793 of whom had a diagnosis of COVID-19. After analyzing admissions, complications, and mortality, they concluded that COVID-19 had a clinical profile different from exacerbations caused by other respiratory viruses during the winter season [17]. Exploring various uses of AI in respiratory health, a particularly interesting 2023 study investigated whether an AI model (specifically ChatGPT) could pass a thoracic surgery board examination [18]. In this case, ChatGPT was given the 150 questions from the competitive exam for the post of Specialist in thoracic surgery announced by the Andalusian Health Service in 2022. The AI correctly answered 58.9% of the questions, with an accuracy rate of 86% for the questions answered correctly [18]. The threshold for passing, as specified in the official exam guidelines, was 60% of the average of the best 10 scores. A passing mark of 40 points was set, and the AI model would have passed this portion of the examination for thoracic surgery physician/specialist with a score of 45.79 points [18].
- Incorporate more recent literature (post-2021) to reflect the current state of AI in oncology and tumor board applications.
We have added a paragraph in the Discussion section with updated references. In 2023, Allowais et al. published a comprehensive review on the use of artificial intelligence (AI) in medicine titled "Revolutionizing healthcare: the role of artificial intelligence in clinical practice" [27]. The authors stated that integrating AI into healthcare holds significant potential for enhancing disease diagnosis, treatment selection, and clinical laboratory testing. AI tools can leverage large datasets and identify patterns that surpass human performance in various healthcare domains. AI offers increased accuracy, cost reduction, and time efficiency while minimizing human errors. It has the potential to revolutionize personalized medicine, optimize medication dosages, enhance population health management, establish clinical guidelines, provide virtual health assistants, support mental health care, improve patient education, and influence patient-physician trust. Despite its promise, AI is still in the early stages of being fully utilized for medical diagnosis. However, growing evidence supports the application of AI in diagnosing various diseases, including cancer [27]. Focusing on the use of AI in lung cancer in recent years, several studies have targeted different stages of the diagnostic and treatment process. For early diagnosis, Wang et al. utilized deep learning to improve low-dose CT (LDCT) screening [28]: A total of 3,326 patients who underwent LDCT for lung cancer screening with subsequent pathological confirmation of lung cancer were included. Their deep learning approach (integrating radiomics and patient demographics) reduced the false-positive rate from 0.41 ± 0.02 to 0.30 ± 0.12, maintained the same false-negative rate, and improved overall diagnostic performance [28]. Similarly, Heidari et al. [29] worked with deep learning (DL) and blockchain technology to address one of the main challenges—preserving data privacy during data sharing. In this study, the authors proposed an approach that utilized a modest amount of data from multiple hospitals and employed blockchain-based Federated Learning (FL) to train a global DL model. Blockchain technology authenticated the data, and FL trained the model on an international scale while maintaining organizational anonymity. The proposed method was trained and tested on datasets including the Cancer Imaging Archive, Kaggle Data Science Bowl, LUNA 16, and a local dataset. The results demonstrated an effective detection of lung cancer patients, achieving an accuracy of 99.69% [29]. To assess treatment response, a systematic review and meta-analysis evaluated the utility of AI in predicting radiotherapy outcomes [30], and in 2024, another study aimed to predict the impact of chemotherapy on cancer cells [31]. The aforementioned meta-analysis [30] included 18 studies with a total of 4,719 patients in which AI models were used to predict survival outcomes in lung cancer patients following radiotherapy. The study demonstrated the clinical feasibility of using AI models to predict post-radiotherapy outcomes in lung cancer patients [30]. Regarding chemotherapy, deep neural network models have been utilized to predict the impact of anticancer drugs on tumors through the half-maximal inhibitory concentration (IC50). These models integrate biological and chemical data to extract relevant features from both genetic profiles and drug compounds. To predict drug response in cancer cell lines, the study employed various deep learning approaches, including Recurrent Neural Networks (RNNs) and Convolutional Neural Networks (CNNs) [31].
- Elaborate on how prompt design could improve AI outcomes. What specific improvements are anticipated?
- Clarify this statement: "Additionally, the prompt should include a current, widely accepted clinical practice guideline across all participating hospitals, and a fine-tuning process should be applied." What does "fine-tuning" entail in this context? Is this a recommendation or part of future work?
Response to comments 4 and 5: We have expanded the Limitations section to include explanations of prompt design and fine-tuning.
Additionally, the prompt should include a current, widely accepted clinical practice guideline across all participating hospitals. Prompt design is a critical factor in optimizing the performance of AI systems like ChatGPT, especially in the context of tumor board decision-making. By carefully structuring prompts, clinicians can ensure that AI outputs are highly relevant, context-specific, and aligned with evidence-based guidelines. For instance, detailed prompts that include patient-specific variables—such as tumor type, stage, genetic mutations, and previous treatment history—can guide the AI to produce recommendations tailored to the clinical scenario. Furthermore, standardizing prompts across cases can enhance consistency, enabling tumor boards to compare AI-generated suggestions more reliably. Effective prompt design also reduces ambiguity, minimizing the risk of irrelevant or incomplete outputs, and fosters the incorporation of up-to-date clinical guidelines by explicitly directing the AI to reference these sources. Overall, improved prompt design has the potential to significantly augment the utility of AI in supporting precise, informed, and reproducible. Prompt design is not the only critical element for enhancing the performance of AI systems like ChatGPT in clinical decision-making; fine-tuning also plays a pivotal role. Fine-tuning involves adjusting a pre-trained model by incorporating domain-specific data, in this case, clinical data related to oncology. This process enables the AI to better understand the nuances of medical cases and generate more precise responses tailored to the clinical context and specific treatment guidelines. For instance, a model fine-tuned with data on different tumor types, genetic variants, and treatment responses could provide more personalized and evidence-based recommendations for real-world patients. Additionally, fine-tuning can integrate updated knowledge from biomedical databases, scientific literature, and authoritative clinical guidelines, such as those from SEPAR, NCCN or ESMO. In the long term, fine-tuning also offers the potential to incorporate local and multicenter datasets to better reflect the characteristics of specific populations—particularly valuable in regions where cancer epidemiology or therapeutic resources differ from global standards. This ensures that AI-generated recommendations are both relevant and applicable to diverse clinical settings. When combined with advances in prompt design and technologies such as federated learning to maintain data privacy, fine-tuning can transform AI into an even more powerful tool, capable of complementing and optimizing tumor board decision-making while improving the quality of care and patient outcomes.
General Comments:
The manuscript is challenging to read due to frequent disruptions in text flow. Reorganize for a clearer, more continuous narrative.
Improve grammar and sentence structure throughout the manuscript for better readability.
Ensure consistency in formatting, including headings, text color, and subheadings.
We have made changes to ensure everything is correct.

Round 2
Reviewer 1 Report
Comments and Suggestions for Authors
Dear Author, I approve the revised version
Reviewer 2 Report
Comments and Suggestions for Authors
Improvements are fully incorporated into the revised manuscript